# Development of Non-Porous Silica Nanoparticles towards Cancer Photo-Theranostics

**DOI:** 10.3390/biomedicines9010073

**Published:** 2021-01-13

**Authors:** Chihiro Mochizuki, Junna Nakamura, Michihiro Nakamura

**Affiliations:** 1Department of Organ Anatomy & Nanomedicine, Graduate School of Medicine, Yamaguchi University, 1-1-1 Minami-Kogushi, Ube, Yamaguchi 755-8505, Japan; mchihiro@yamaguchi-u.ac.jp (C.M.); jnakam@yamaguchi-u.ac.jp (J.N.); 2Core Clusters for Research Initiatives of Yamaguchi University, 1-1-1 Minami-Kogushi, Ube, Yamaguchi 755-8505, Japan

**Keywords:** silica nanoparticles, organically modified silica (ORMOSIL) nanoparticles, organosilica nanoparticles, imaging, therapy, theranostics, nonporous

## Abstract

Nanoparticles have demonstrated several advantages for biomedical applications, including for the development of multifunctional agents as innovative medicine. Silica nanoparticles hold a special position among the various types of functional nanoparticles, due to their unique structural and functional properties. The recent development of silica nanoparticles has led to a new trend in light-based nanomedicines. The application of light provides many advantages for in vivo imaging and therapy of certain diseases, including cancer. Mesoporous and non-porous silica nanoparticles have high potential for light-based nanomedicine. Each silica nanoparticle has a unique structure, which incorporates various functions to utilize optical properties. Such advantages enable silica nanoparticles to perform powerful and advanced optical imaging, from the in vivo level to the nano and micro levels, using not only visible light but also near-infrared light. Furthermore, applications such as photodynamic therapy, in which a lesion site is specifically irradiated with light to treat it, have also been advancing. Silica nanoparticles have shown the potential to play important roles in the integration of light-based diagnostics and therapeutics, termed “photo-theranostics”. Here, we review the recent development and progress of non-porous silica nanoparticles toward cancer “photo-theranostics”.

## 1. Introduction

Nanomedicine has made great strides as an interdisciplinary research field that combines various scientific fields, such as nanotechnology, organic chemistry, materials science, quantum science, molecular biology, and biotechnology, for biomedical applications [1,2,3,4,5,6,7,8,9,10,11,12,13,14,15,16,17,18,19,20,21,22,23]. Recent studies have revealed that nanoparticles have excellent potential for the creation of innovative medicines. Applications of nanoparticles in the biomedical field have various advantages, such as numerical superiority, multifunctionality, additive/multiplier effects, and nano-sized effect, as compared with the small molecules used as therapeutics and imaging contrast agents. Various functional nanoparticles, such as liposomes, dendrimers, gold nanoparticles, iron oxide, and silica nanoparticles, have been developed and investigated in the biomedical field [24,25,26,27,28,29,30,31]. In biomedical imaging in particular, nanoparticles have been applied to various imaging modalities, including optical imaging, magnetic resonance imaging (MRI), X-ray computed tomography (CT), and radiation imaging. Each of these image modalities has its advantages and disadvantages. Optical imaging has high sensitivity—comparable to that of radiation imaging—and high resolution, compared to other modalities. In addition to microscopic imaging using visible light, which is widely used at present, optical imaging has expanded to in vivo imaging by applying near-infrared light. Applications of silica nanoparticles for optical imaging are also actively progressing. Unlike quantum dots or iron oxide nanoparticles, silica nanoparticles do not possess an inherent imaging signal; however, they can be effectively multi-functionalized using various functional materials. In addition to imaging, the application of optics is also being expanded to therapeutic techniques, such as photodynamic therapy (PDT). PDT is a breakthrough medical technique which irradiates only the lesion area with an optical laser, in order to treat cancer without resorting to surgery, further reducing complications [32,33]. Recent research in nanomedicine has been progressing toward the realization of the integration of diagnosis and therapy, termed “theranostics”. The integration of optical imaging and phototherapy, which can be termed “photo-theranostics”, is one of the most important nanomedical goals at present. Many reviews focused on imaging and therapy, as well as “theranostics”, using functional nanoparticles have been published, considering various diseases such as cancer and neurodegenerative [34,35], cardiovascular [36,37], and autoimmune (particularly rheumatoid arthritis) [38,39] diseases [11]. Cancer “theranostics” is one of the most active fields in nanomedicine. Mesoporous silica nanoparticles have been researched and well-reported for the development of cancer “theranostics” by providing excellent drug delivery systems [40,41,42,43,44,45,46,47,48]. However, few reviews focus on these aspects using non-porous silica nanoparticles [18,49,50,51]. In this review, we discuss the recent development and progress of non-porous silica nanoparticles towards cancer “photo-theranostics”.

## 2. Silica Nanoparticles

In 1968, Stöber et al. proposed a pioneering method for the synthesis of silica nanoparticles which are spherical and monodisperse from aqueous alcohol solutions of alkoxysilane with the presence of ammonia as a catalyst [52]. In this method, the silica nanoparticles are synthesized by the hydrolysis and condensation of alkoxysilane. Subsequently, other synthesis routes have also been discovered for the preparation of silica nanoparticles of different shapes and sizes. The reverse microemulsion method conveniently synthesizes the nanoparticles in a reverse water-in-oil (W/O) microemulsion. The silica nanoparticles are grown inside the microcavities by controlling the addition of silicon alkoxides and catalyst during the process [53,54,55].

The alkoxysilanes, which are precursors of silica nanoparticles, contain carbon; however, the formed silica nanoparticles do not contain any carbon, due to hydrolysis. Therefore, these silica nanoparticles could be termed “inorganosilica nanoparticles”. Another type of silica nanoparticles has been prepared by various precursors. These silica nanoparticles are referred to herein as “organically modified silane (ORMOSIL) nanoparticles” and “functional organosilica nanoparticles”. These novel silica nanoparticles contain exposed interior and surface carbon, as well as organic and functional groups (Figure 1). ORMOSIL nanoparticles are defined as those which contain carbonaceous moieties (e.g., octyl and vinyl groups) [56,57] and do not contain any functional residues (e.g., amines or thiols). Functional organosilica nanoparticles contain functional groups such as thiol and epoxy groups [58,59,60,61]. Functional organosilica nanoparticles can be prepared from an organosilane reagent containing functional residues, such as (3-mercaptopropyl)trimethoxysilane (MPMS) or 2-(3,4-epoxycyclohexyl)ethyltrimethoxysilane (EpoMS). The prepared organosilica nanoparticles contain the functional residues, while chain alkanes originate from silane coupling agents in the interior and on the surface.

The application of silica nanoparticles in the biomedical field can provide great advantages. Specially size-controlled nanoparticles have useful behavior in the body, compared to small molecules. Small molecules disperse to different organs, tissues, cells, and intracellular organelles in the body. On the other hand, nanoparticles have relatively long-term retention time in the body, as appropriately sized nanoparticles can accumulate in tumor tissue significantly better than in ordinary tissues; this phenomenon is known as the enhanced permeability and retention (EPR) effect [62,63]. Silica particles of various sizes with a narrow size distribution can be prepared, which have been applied to study nano-sized effects. The appropriate size of nanoparticles in biodistribution for cancer therapy is currently an ongoing discussion. Furthermore, the degradability and clearance of nanoparticles are essential phenomena for biomedical uses. Both biodegradable and renal clearable nanoparticles are known to cross the globular filtration obstruction of the kidneys, in order to be securely discharged. Silica nanoparticles have a fast hydrolytic rate and dissolve, over time, into water-soluble silicic acid, which is excreted in urine [64,65]. The toxicity of the silica particles in in vivo systems has also been examined systematically [66,67]. The acute and sub-chronic toxicities of 50 and 500 nm non-porous silica nanoparticles and 500 nm mesoporous silica nanoparticles have been investigated by single-dose intravenous injection, where 50 nm non-porous silica nanoparticles and 500 nm mesoporous silica nanoparticles showed less sub-chronic toxicity [68].

Silica particles are excellent, due to the ability to control the morphology, particle size, uniformity, and dispersity; furthermore, they have been shown to be safe for biological applications. Silica nanoparticles combined with functional materials also present unique functions and structures. In the following section, we describe the characteristics and potential for multifunctionalization of non-porous silica nanoparticles, including synthesis methods.

### 2.1. Inorganosilica Nanoparticle

The synthesis of inorganosilica particles involves the hydrolysis and condensation of metal alkoxides (Si(OR)_4_), such as tetraethylorthosilicate (TEOS, Si(OC_2_H_5_)_4_) or tetramethoxyorthosilicate (TMOS, Si(OCH_3_)_4_), in the presence of a mineral acid (e.g., HCl) or base (e.g., NH_3_) as the catalyst [55,69]. Inorganosilica nanoparticles are composed of an internal siloxane bridge (-Si-O-Si-) and possess silanol groups on their surfaces. Stöber et al. first developed an effective method for synthesizing monodispersed silica particles, which led to the controlled synthesis of submicron-sized spherical silica nanoparticles (5–2000 nm) [52]. The main advantage of this method is its ability to synthesize monodispersed spherical silica particles, compared to the acid-catalyzed systems [55]. From this discovery, mono-dispersed inorganosilica nanoparticles have been synthesized from metal alkoxides with ammonia as catalyst, using the sol-gel process named the Stöber method.

### 2.2. ORMOSIL

Schmidt et al. and Wilkes et al. reported the chemical modification of siloxanes, with the results called “ormocers” [70,71] and “ceramers” [72]. The chemical reactivities of siloxanes offer a wide range of possibilities for chemical modification of the molecular precursors. After that, the organic modification of silane to nanoparticles such as vinyl triethoxysilane (VTES) and octyl triethoxysilane (OTES) [57] has been widely used and named ORMOSIL. In the synthesis procedure, VTES readily aggregate under reverse micelles condition, where the triethoxysilane moieties are hydrolyzed to form the silica network. As-prepared ORMOSIL nanoparticles possess vinyl groups on the surface, toward the hydrophobic side of the micellar interface. The surface vinyl groups of ORMOSIL nanoparticles have been converted to other functional groups, such as carboxylic groups [57]. ORMOSIL nanoparticles synthesized from two or more varieties of organosilicates have been prepared. ORMOSIL nanoparticles with amino functional group were synthesized by synchronous hydrolysis of VTES and 3-aminopropyltriethoxysilane (APS). These ORMOSIL nanoparticles possess functional groups on the surface, as prepared.

### 2.3. Functional Organosilica Nanoparticles

Thiol-organosilica nanoparticles have been successfully prepared from thiol-organosilicate using a one-pot synthesis [60]. The synthesis processes of thiol-organosilica nanoparticles has been compared for three kinds of organosilicates: MPMS, MPES ((3-mercaptopropyl)triethoxysilane) and MPDMS ((3-mercaptopropyl)methyldimethoxysilane). The formation trends and rates for these thiol-organosilica nanoparticles varied with differing precursors, concentrations, and synthetic conditions. The Stöber method is not suitable for the formation of thiol-organosilica nanoparticles, as compared with the case of TEOS. However, the conditions of 27% ammonium hydroxide without ethanol are suitable for the formation of thiol-organosilica nanoparticles. The mean diameters of nanoparticles increase along with the increase in concentrations of silicate, while the size distributions of nanoparticles prepared under various conditions also differ between silicate sources. Epoxy-organosilica nanoparticles were prepared from 2-(3,4-epoxycyclohexyl)ethyltrimethoxysilane (EpoMS) as a single silica source in a one-pot synthesis method. The mean diameters of the particles depended on the EpoMS concentration with a narrow coefficient of variation (5.9%). The surfaces of the particles had unique properties, such as positive zeta potential. The positively charged surface allows these particles to bind to DNA and proteins [61]. These functional organosilica nanoparticles were prepared using a single silicate source. Functional organosilica nanoparticles can be classified as either “single-silicate” or “multi-silicate”, based on the number of organosilicate sources [18]. At present, multi-silicate types of functional organosilica nanoparticles are being developed, due to the potential for further additional functions, such as biodegradability [73].

## 3. Multi-Functionalized Silica Nanoparticles

Multi-functionalization of silica nanoparticles is very important for their biomedical application. For light applications, silica nanoparticles are commonly functionalized with fluorescence materials, such as organic fluorescent dye. Silica nanoparticles are expected to be applied to advanced technologies, such as multimodal imaging and theranostics. These advanced technologies require functional nanoparticles for further multifunctionality. Therefore, in this section, we describe the functional nanoparticles used to functionalize silica nanoparticles, in addition to fluorescent materials.

### 3.1. Fluorescence Materials

Fluorescent silica nanoparticles have been widely used in biological applications for bioassay [74,75,76] and imaging, due to their flexible chemistry and apparent biocompatibility. In this review, we focus on their use for optical imaging. Optical imaging has been used for microscopic observations for many decades, due to the high spatial resolution and exceptional detection sensitivity. Fluorescence imaging is one of the most widely used optical imaging techniques, which allows for visualization of the biomolecules, gene expression, and enzyme activity in living cells and tissues, in order to understand their biological functions. Novel nanoscale fluorescent materials are integral to emergent fields, such as nanobiotechnology, and facilitate new research in various contexts. Silica nanoparticles are excellent host materials for fluorescent probes, which are fluorescent organic and inorganic materials (see Table 1). Fluorescent silica nanoparticles have shown tremendous promise as optical fluorescence probes for biological imaging [77,78].

#### 3.1.1. Organic Materials

The most commonly used optical imaging probes are small organic molecules. Synthetic organic fluorescent dyes, such as fluorescein, were the first fluorescent compounds used in the history of biological research. Various fluorescein derivatives have been developed, in order to improve their photostability and solubility for biological applications. Preparations of monodisperse fluorescein isothiocyanate (FTIC)-labeled silica nanoparticles by the Stöber method were reported in 1992 [81,82]. Blaaderen et al. described a general synthetic procedure to incorporate FITC, through covalent bonds, into monodisperse colloidal silica spheres. The process consists of the dye coupling to a silane coupling agent, (3-aminopropyl)triethoxysilane, facilitating the controllable incorporation of the reaction product into the silica sphere. The position of incorporation of FITC in the silica particle could also be controlled [81]. Rhodamine B has been encapsulated in organosilica nanoparticles, prepared in a one-pot synthesis using a direct method [60,61]. The direct method was applied to inorganosilica nanoparticles with rhodamine B, in order to compare the incorporation efficiency of fluorescence molecules of organosilica nanoparticles, but no fluorescence was observed from the inorganic nanoparticles prepared from TEOS. Rhodamine B can interact with the internal structure of thiol-organosilica through hydrophobic and electrostatic interactions, due to the presence of unique internal structures, such as carbonaceous moieties and functional residues of thiol-organosilica. To evaluate the direct method, fluorescence thiol-organosilica nanoparticles were prepared by the direct method [60] and the conjugation method [83]. Fluorescent organosilica nanoparticles prepared by the direct method possessed higher fluorescence intensities than those prepared by conjugation method. Ow et al. described highly fluorescent and photostable core–shell nanoparticles using a modified Stöber synthesis, with a size range of 20–30 nm. These nanoparticles were 20 times brighter than their constituent fluorophore [103]. Yari et al. described yellowish eosin, which is the form of eosin used most often as a histologic stain, as a fluorophore in a peroxyoxalate chemiluminescence system using silica nanoparticles [84]. Cyanine dyes are cationic molecules in which two heterocyclic units are joined by a polyene chain [85]. Cy3, Cy5, indocyanine green (ICG), IRdye 800CW, and IR820 are well-known fluorescence probes for bioimaging [74,79,86,87]. Among them, ICG is an FDA-approved near-infrared (NIR) fluorescent dye used in clinical imaging;. IRdye 800CW is currently undergoing preclinical trials. ICG molecules ion-paired with a cationic polymer polyethylenimine (PEI) have been encapsulated into a silica nanoparticle using the Stöber method. PEI acted to prevent the aggregation of ICG in the silica and reduced the self-quenching of fluorescence [79]. IR820 can be used as a near-infrared (NIR) fluorescence imaging agent. Various IR820-grafted silica nanoparticles have been synthesized using several methods and applied to NIR fluorescence imaging in vivo [104,105,106].

#### 3.1.2. Inorganic Materials

Interest in the development of rare earth-based nanoparticles has increased, whose optical properties and low cytotoxicity are promising for biological applications. The unique properties of rare earth-based nanoparticles include their high photostability, absence of blinking, narrow emission lines, large Stokes shifts, and long lifetimes; thus, they can be well-exploited for bioimaging applications [107]. Their luminescence properties make them good optical probes for silica nanoparticles [88,89,90].

Eu^3+^-doped silica nanoshell particles with 100 and 200 nm diameters have been synthesized using the sol-gel reaction. The Eu^3+^ doped silica nanoshells exhibited a narrow red emission line at 615 nm by UV excitation. The long lifetimes of rare earth ions have facilitated studies using two-photon microscopy [91]. Ruthenium complex-doped silica nanoparticles are some of the most commonly used for fluorescence imaging or as photostable biomarkers [92,93,94]. Bagwe et al. reported the formation of tris(2,2′-bipyridyl) dichlororuthenium(II) (Ru(bpy)_3_^2+^) dye-doped silica nanoparticles by ammonia-catalyzed hydrolysis of tetraethyl orthosilicate (TEOS) in a water-in-oil microemulsion. Its emission spectrum was more red-shifted in the presence of an anionic surfactant. The Ru(bpy)_3_^2+^ dye molecule is sensitive to the chemical environment around the dye molecule [95]. Rossi et al. developed a simple method to prepare bright and photostable luminescent silica nanoparticles of different sizes with narrow size distribution in high yield (80%). The silica nanoparticles contained the transition metal complex tris(1,10-phenanthroline) ruthenium (II) chloride([Ru(phen)_3_]Cl_2_). The method was based on the use of Stöber synthesis in the presence of Ru(phen)_3_^2+^—a ruthenium diimine complex—to form bright silica particles. They demonstrated that digital counting of the luminescent nanoparticles could be used as an alternative to detection techniques involving analog luminescence detection in bioanalytical assays [96]. Ru-doped silica nanoparticles were used in an electrogenerated chemiluminescence (ECL) detection system [97,98]. Quantum dots (QDs) possess several advantages over traditional organic dyes, such as broad excitation spectra, size-dependent fluorescence properties, long emission times, and photostability. Silane encapsulation of various metals and QDs has been reported, in order to maintain their properties and function. QDs encapsulated with a thiol-organosilica layer (thiol-OS-QDs) possessed unique photophysical properties, through encapsulation within the thiol-organosilica layer containing organic dye [99].

### 3.2. Nanoparticles

Multifunctional silica nanoparticles have been designed to make use of the properties of combined materials, i.e. organic molecules or inorganic materials such as novel metals and metal oxides. Among them, the multifunctionalization of silica nanoparticles with functional nanoparticles, such as gold nanoparticles or iron oxide nanoparticles, has garnered great attention for given biomedical applications. Multifunctional silica nanoparticles provide a significant impact not only in the biomedical field, but also for physical chemistry (e.g., for spectroscopy and magnetism). In these multifunctional nanoparticles, silica particles are used as cores or shells. In this section, we describe multifunctional silica nanoparticles, based on their structure.

#### 3.2.1. Silica Shell Nanoparticles

Silica as a coating material has progressed to an unparalleled application for improving colloidal properties and abilities using core–shell rational designs, as well as benefitting from its manufacturing flexibility [108]. Several reports have focused on the silica coating of inorganic nanoparticles by aqueous methods, such as the Stöber method [109,110], silane coupling agents, and sodium silicate; however, recently, water-in-oil (W/O) microemulsions [111] and the coating of polymeric aggregates with silica [112,113] have also been reported. Tuning of the relaxometry of γ-Fe_2_O_3_@SiO_2_ core–shell nanoparticles was carried out by changing the thickness of the covering silica layer. The control of silica shell thickness can, accordingly, be tuned to allow a high response of the contrast agent [108].

#### 3.2.2. Silica Core Nanoparticles

Silica nanoparticles fabricated from inorganic nanoparticles [114,115,116,117,118,119,120,121] provide attractive structures, due to the exposed inorganic nanoparticles which may facilitate magnetism and catalysis [118,119]. Monodisperse SiO_2_ nanoparticles (230 nm) were synthesized by the Stöber method and used as the core for Fe_2_O_3_ catalysts through hydrolysis precipitation. Fe_2_O_3_ nanoparticles were deposited to the silica surface through a hydroxyl bond [118]. Furthermore, gold nanoshells with a silica core have shown unique optical properties. The observed plasmon band in the NIR region relies on their center range and shell thickness by multipolar plasmon resonances [114,115,116]. The optical properties in the NIR region are considered to assess their potential for bioapplication in the biological window, as living cells and tissues have low light scattering and absorption in the NIR region. In particular, photothermal effects in the NIR region have gained attention for photo hyperthermia therapy, as discussed below.

## 4. Optical Imaging

Notable advantages of fluorescent imaging techniques include their high sensitivity, high temporal resolution, multichannel imaging using multiple fluorescent probes, and low cost. Fluorescent nanoparticles are promising for fluorescent microscopic imaging, including cellular, in vivo, and multimodal imaging, due to their high fluorescence intensity and photostability.

### 4.1. Cellular Imaging

#### 4.1.1. Targeted Cell Imaging

Aptamer-conjugated inorganosilica has been used in a range of applications; particularly, dual nanoparticles have been used for magnetic extraction and fluorescent labeling on CCRF-CEM cells (CCL-119 T-cell, human acute lymphoblastic leukemia), Ramos cells (CRL-1596, B-cell, human Burkitt’s lymphoma), and MCF-7 (human breast carcinoma) [122,123,124]. Rhodamine B-contained florescence ORMOSIL has been conjugated with various bioactive molecules, such as transferrin and monoclonal antibodies (e.g., anti-claudin 4 and anti-mesothelin), for targeted delivery to pancreatic cancer cell lines. The cellular uptake of these bioconjugated (targeted) nanoparticles was significantly higher than that of non-conjugated ones [125]. QDs have been extensively used, throughout the past decade, as potential luminescence markers for cell labeling in biological applications [99,100,101], since they possess the unique optical property of an emission spectrum which is tunable with their size and shape. Dual-fluorescent thiol-organosilica nanoparticles composed of QDs and fluoresceine in a thiol-organosilica layer were synthesized by a one-pot process for biological application. The dual-fluorescent thiol-organosilica nanoparticles showed reduced photoblinking, due to the nature of QDs. Cells labeled with dual-fluorescent thiol-OS-QDs have been functionalized with an antibody against CD20 using the thiol residue on the surface, which was useful for the molecular tracing of tumor antigen on Raji cell (B lymphoblastic cell line) [99].

#### 4.1.2. Functional Imaging

The interactions between macrophages and nanoparticles are vital for nanomedicine developments, as macrophages uptake the greater part of nanoparticles in vivo. Imaging systems for interactions between macrophage and fluorescent thiol-organosilica nanoparticles are often an essential model. Fluorescent inorganosilica nanoparticles have been applied to cellular imaging. Multifluorescent silica nanoparticles have been synthesized, by the Stöber method, using conjugates of (3-aminopropyl)triethoxysilane and fluorescent dye NHS esters. Additionally, multifluorescent silica nanoparticles containing two types of fluorescent dyes have been synthesized and utilized in biological applications. Microscopy analysis has shown high and tuned fluorescence and multiple fluorescences from single nanoparticles and cells labeled with multifluorescent organosilica nanoparticles. The organosilica nanoparticles exhibited no toxicity on in vivo injection into labeled cells [83]. Fluorescent organosilica nanoparticles with 100 nm diameter have been applied to time-lapse fluorescence imaging. Time-lapse fluorescence imaging of mouse peritoneal macrophages showed cellular uptake, while single-cell analysis also showed various patterns of uptake kinetics, which were quantitatively evaluated. A correlation between morphologic findings and endosomal uptake over time was also observed and quantitatively analyzed. Fluorescent organosilica nanoparticles have shown high potential as fluorescence markers for time-lapse fluorescence imaging and single quantitative cell functional analysis for nanomedicine development [126].

An in vitro imaging system to evaluate the stealth function of nanoparticles has been established using fluorescent organosilica nanoparticles [127]. Surface-functionalized organosilica nanoparticles with polyethylene glycol (PEG) were prepared, possessing a brush-type PEG layer. Time-lapse fluorescence imaging of mouse peritoneal macrophages by fluorescent thiol-organosilica nanoparticles demonstrated their cellular uptake. The single-cell quantitative analysis showed various patterns of uptake kinetics, indicating the heterogeneous functionality of macrophages. The cellar uptake level and kinetics of each macrophage were different, depending on the surface structure of the thiol-organosilica nanoparticles. A dual-particle administration study simulated the cellular uptake of nanoparticles possessing different surface structures. Macrophages incubated with Flu-PEG30K and Rho showed various cellular uptake patterns. The cellar uptakes of Flu-PEG30K and Rho into macrophage cells (Figure 2) were different. Many cells showed uptake of only Rho, while some cells showed uptake of both Rho and Flu-PEG30K. Many cells showed only Rho fluorescence (B1 and B2), and few cells showed fluorescence of both Flu and Rho (B3–B6). Single-cell imaging and analysis revealed various cellular uptake patterns and kinetics of bare and PEGylated nanoparticles. The PEGylated nanoparticles possessed a stealth function against most macrophages (i.e., PEG-sensitive macrophages). The authors proposed that PEG-insensitive macrophages, including PEG-resistant macrophages, are a novel concept. This concept is essential for understanding and regulating the immune responses against PEGylated nanoparticles. The stealth function against specific macrophages (i.e., PEG-insensitive macrophages) is under investigation.

As-synthesized highly bright and photostable cyanine dye-doped inorganosilica nanoparticles (IRIS Dots) [128] efficiently labeled human mesenchymal stem cells (hMSCs). Moreover, non-functionalized IRIS dots allow for discrimination between live and early-stage apoptotic stem cells through a definite external cell surface distribution, compared with FITC-labeled Annexin V probes [86]. The cell uptake of ORMOSIL nanoparticle processes has been characterized by the mechanism underlying endocytosis and the sub-cellular localization of ORMOSIL nanoparticles for A539 cells. A better understanding of the mechanism could provide an improved understanding of the potential toxicity, as well as better applications of ORMOSIL nanoparticles in biomedicine [129].

### 4.2. In Vivo Imaging

For in vivo fluorescence imaging, much attention has been paid to high sensitivity with a sensitive camera, in order to detect the fluorescence emission from fluorophores in whole-body living small animals. In particular, fluorescent silica nanoparticles possess high brightness and stability for the observation conditions. Furthermore, fluorescence imaging for cells and tissue in the NIR wavelengths between 700 and 900 nm has an advantage for in vivo imaging, due to the low absorption of biological molecules in this region (NIR window). The development of probes that combine low toxicity with high sensitivity, resolution, and stability has strongly accelerated in vivo imaging [102,103,104,106,130,131,132,133,134,135].

#### 4.2.1. Fluorescence Imaging

In 2005, Wiesner et al. proposed ultrasmall silica nanoparticles, which possess highly fluorescent core–shell silica with a narrow size distribution [103]. The fluorescent silica nanoparticle later became known as Cornell dots (C dots); however, these C dots have not been shown to be non-toxic at biologically relevant concentrations. They can be used in wide range of imaging applications, including sentinel lymph node mapping, intravital visualization of capillaries and macrophages, and peptide-mediated multicolor cell labeling for real-time imaging of tumor metastasis and tracking with visible light for tetramethylrhodamine (TRITC) dye [130]. NIR fluorescence imaging can be used for biomedical development [131,132,133]; however, some NIR fluorescent probes—including fluorescent dye molecules and semiconductor QDs—are known to be toxic. Highly luminescent and monodisperse semiconductor QDs (Si@QDs@Si nanoparticles) have been synthesized [102,135]. The water-compatible Si@QDs@Si exhibited low toxicity and 200-fold stronger photoluminescent emission than single QDs. The highly amplified fluorescence signals from Si@QDs@Si-NP-tagged cells could be observed for 10 days in vivo. The detection of fluorescence of Si@mQDs@Si-NP-labeled cells was achieved with as few as 400 nanoparticles in mouse skin [102].

#### 4.2.2. Tumor Imaging

Controlling the biodistribution of nanoparticles is key to achieving target specificity for theranostics [136,137]. In particular, the nanoparticle size affects the biodistribution of nanoparticles throughout the body [138,139]. One way to achieve selective nanoparticle transportation to tumors is to exploit tumor vasculature abnormalities; namely, hypervascularization, aberrant vascular architectures, extensive production of vascular permeability factors stimulating extravasation within tumor tissues, and lack of lymphatic drainage. The nanoparticles selectively extravasate in tumor tissues, due to their abnormal vascular nature. This phenomenon is known as the enhanced permeability and retention (EPR) effect [62,63]. Maeda and Matsumura first discovered the EPR effect mechanism of tumoritropic accumulation of smancs and other proteins. To determine the general mechanism, they used radioactive (^51^Cr-labeled) proteins of various molecular sizes (Mr 12,000–160,000), as well as other properties and dye-complexed serum albumin, to visualize the accumulation in tumors of tumor-bearing mice [62]. From this discovery, the EPR effect became the basic strategy for nanoparticle delivery by remodeling the tumor microenvironment based on tumor vasculature targeting. Qian et al. reported the synthesis of ORMOSIL nanoparticles—which encapsulated either PpIX (protoporphyrin IX) photosensitizers or IR-820 fluorophores—and their use for direct excitation of the fluorescence of PpIX by two-photon absorption in the intracellular environment of tumor cells, as well as their cytotoxicity towards tumor cells. IR-820-doped ORMOSIL nanoparticles have been applied to the in vivo brain imaging of mice and their application to sentinel lymph node (SLN) mapping of mice has been demonstrated. They also demonstrated that intravenously injected NIR nanoparticles could target the subcutaneously xenografted tumor of a mouse through blood circulation and the EPR effect [104]. Near-infrared fluorescent silica–porphyrin hybrid nanotubes (HNTs) have been successfully synthesized by π–π stacking and electrostatic interaction through a sol-gel reaction. The HNTs did not show toxic liability in vitro or in vivo. The HNT-labeled macrophages were detected in vivo, where even 200 labeled cells were detectable. Furthermore, the biodistribution of HNT-labeled macrophages has been tracked by fluorescence imaging [131]. Aminocyanine dye-encapsulated silica nanoparticles were synthesized using a reverse microemulsion method. The brightness and photostability of covalent dye-linked fluorescence nanoparticles correspond to the anchoring sites of the aminocyanine dyes. The brightness and photostability of fluorescence nanoparticles with NIR emission indicate their potential for long-term and real-time bioimaging applications [132].

Near-infrared fluorescence and organosilica nanoparticles (thiol-OS/IR820) have been prepared using a thiol organosilica compound and IR-820, which is a cyanine-based near-infrared dye. IR-820 was mainly present inside the particles and changed their optical properties. Multiple new fluorescence peaks appeared, where the fluorescence occurred in a wide range of the NIR region. Furthermore, the up-conversion phenomenon was also observed (a phenomenon in which low-energy light is converted into high-energy light), occurring due to the structure of the organosilica nanoparticles. The fluorescence bioimaging of subcutaneous xenograft tumor mice using thiol-OS/IR820 has been performed. Thiol-OS/IR820 accumulated in the cancer tissue due to the EPR effect with intravenous administration, where the fluorescence was observed in a depth-dependent manner (Figure 3). With short-wavelength excitation light, the cancer tissue in the shallow part near the surface was detected with high sensitivity. Moreover, with the longer wavelength of the excitation light, the cancer tissue in the more in-depth part could be observed, while macrophages which had taken up thiol-OS/IR820 in the liver and spleen were also detected. Thiol-OS/IR820 has shown low toxicity in vivo and fluorescence bioimaging using it enables the long-term observation of cell dynamics, thus providing an excellent possibility for the visualization and discovery of new biological phenomena in vivo [106].

### 4.3. Multimodal Imaging

Functionalized silica nanoparticles present remarkable properties that are not found in bulk materials. Integrating several kinds of materials at the nanoscale is critical to the success of multimodal imaging nano devices. Various types of in vivo imaging techniques are available, any of which have certain advantages and limitations, in terms of sensitivity, signal attenuation, and resolution. The combination of several in vivo imaging modalities has shown potential for theranostic image-guided drug delivery, including optical imaging, magnetic resonance imaging (MRI), and nuclear imaging, such as X-ray computed tomography (CT) and positron emission tomographic (PET). These modalities are complementary, due to their different properties, while multimodality approaches are applied to overcome the limitations of a single modality [140].

#### 4.3.1. Optical-Magnetic Resonance (MR) Imaging

Multimodal imaging combining optical imaging (especially fluorescent imaging) and MRI has unique relations, as the advantages and disadvantages of the separate methods compensate for each other. MR imaging is readily utilized in T1- and T2-weighted contrasts and has several advantages, such as higher spatial resolution and the fact that physiological, molecular, and anatomical information can be extracted simultaneously. However, the sensitivity of MRI is not high and, thus, a long scan time and a large quantity of probes are needed. On the other hand, optical fluoresce imaging has been developed for in vitro and in vivo imaging applications. The advantages of fluorescence imaging are high temporal resolution, high sensitivity, multichannel imaging with multiple fluorescent probes, and relatively low cost. However, fluoresce imaging is not quantitative, and the image information is a surface-weighted in vivo system. Thus, multimodal imaging utilizing a combination of MRI and fluorescent imaging could facilitate the creation of a novel imaging system [141,142,143].

Silica can act as a coating material for a magnetic nanoparticle core [144,145,146,147] and can also be doped with Gd^3+^ or Mn^2+^ ions [148,149] to render the material MRI-active for use with MR contrast agents. The coating of nanoparticles of other materials with silica is one of the most popular strategies for developing multifunctional silica. Hydrophobic iron oxide cores can be coated with silica shells using the reverse microemulsion method or siloxanes which contain hydrophobic parts, thus resulting in an MRI-detectable silica nanoparticle. Li et al. reported Fe_3_O_4_@SiO_2_@SLCONHR nanoparticles by integrating highly Aβ-specific and turn-on fluorescence cyanine sensors for a neuroprotective dual-modal nanoprobe in vivo for both NIR imaging and MRI contrast [143]. Inorganic nanoparticles, such as Fe, are readily utilized as MRI biomedical imaging probes, due to their inherent T1- and T2-weighted contrasts. The nanoprobe thus must integrate highly Amyloid-β peptide (Aβ)-specific and turn-on fluorescence cyanine sensors. Aβ is known as a molecule which can cause Alzheimer’s disease (AD) through its accumulation and deposition within the frontal cortex and hippocampus in the brain. The Aβ-specific nanoprobe is not non-toxic and non-invasive, but it has high permeability of the blood–brain-barrier (BBB, a tight junction of endothelial cells), as demonstrated by fluoresce imaging and MRI. Fe_3_O_4_@SiO_2_@SLCONHR has been injected into transgenic (Tg) mouse overexpressing Aβ species, in order to observe the MR signal in the brain for evaluation as a T2 contrast agent for imaging Aβ species in vivo. The bright MR signal turned dark after the injection of Fe_3_O_4_@SiO_2_@SLCONHR. This indicates that many nanoparticles crossed the BBB and accumulated in the brain. Furthermore, a quantitative analysis of the relaxation times revealed that the mean signal intensity in Tg mice decreased ≈30% after 2 h post-injection, with a signal recovery occurring 4 h post-injection. The longer retention of nanoparticles in Tg mice revealed by MRI in vivo was according to that observed in fluorescence imaging in vivo, indicating that the presence of Aβ species in Tg mice could retard the washout of nanoparticles from the brain, due to the strong interactions of Fe_3_O_4_@SiO_2_@SLCONHR with Aβ species. Ex vivo fluorescence images of brain slices revealed the specific colocalization of Fe_3_O_4_@SiO_2_@SLCONHR with the Aβ plaque in detail. Then, the mice were scanned for high-resolution ex vivo MRI without physiological motion artifacts. Many dark spots were found in different regions of Tg brains, affording disease-specific MR signals. The nanoparticles were successfully applied for in vivo fluorescence imaging with high sensitivity and selectivity to Aβ species, as well as MRI with high spatial resolution in Tg mice. The nanoparticles showed potential as a powerful in vivo dual-modal imaging tool for the early detection and diagnosis of AD in humans.

#### 4.3.2. Optical-X ray Computed Tomography (CT) Imaging

X-ray CT is one of the most widely used diagnostic imaging modalities, which provides anatomical information including the location, shape, and size of tissues. However, the disadvantage of X-ray CT is its low contrast for soft tissue. In clinical use, the X-ray CT contrast agents which are most often used are iodine-containing molecules, used only for angiography and urography due to their non-specificity to tissues. Moreover, the major limitations of X-ray use are a lack of molecular specificity and a risk of carcinogenesis. On the other hand, optical photon energies provide intrinsic molecular specificity by comparison with molecular energy levels. Therefore, the combination of X-ray CT and optical imaging—especially fluorescence imaging—provides one of the most effective combinations. Functionalized silica nanoparticles which contain CT contrast agents, such as I and Au, can be used as probes for CT–fluorescence dual-mode imaging in vivo [80,150,151,152,153]. This could help to decide on a therapeutic strategy before surgery and to identify the tumor during surgery.

NIR fluorescent silica-coated gold nanoparticle clusters (Au@SiO_2_) with high X-ray absorption coefficient were applied to multimodal optical-X-ray CT imaging for lymph nodes (LNs) and lymph vessels (LVs). The visualization of LNs and LVs is vital for preventing the spread of cancer or when operating on lymphedema. Au@SiO_2_ nanoparticles were synthesized by hydrolysis condensation of a tetrakis(4-carboxyphenyl)porphyrin (TCPP)-binding, which possessed NIR fluorescence, causing a red-shift of the absorption band by π–π stacking of TCPP with a silica precursor in the presence of gold nanoparticles at room temperature. The Au@SiO_2_ nanoparticles provided stronger CT contrast than Iopamiron (a state-of-the-art CT contrast agent) and enabled visualization of the LNs and the LVs by CT, thus providing accurate anatomical information, location, and size. Interestingly, the Au@SiO_2_ nanoparticles clearly visualized LNs and LVs by fluorescence imaging. As a result, the LNs were successfully extracted, which were not identified with the naked eye [154].

#### 4.3.3. Optical-Positron Emission Tomographic (PET) Imaging

Optical medical imaging has limited penetration and small field of view. PET is based on the detection of high-energy photon pairs produced during an annihilation collision between a positron and an electron, from which 3D images can be reconstructed by computer analysis. PET contrast agents contain positron-emitting radionuclides, such as ^11^C, ^18^F, ^64^Cu, ^68^Ga, ^89^Zr, and ^124^I. The notable advantages of PET include unlimited depth of penetration and excellent sensitivity. The disadvantage of PET imaging is its low spatial resolution. This low spatial resolution can be compensated by intraoperative optical fluorescence imaging in dual-modality fluoresce imaging/PET imaging methods. Multimodal concepts help to combine the complementary strengths of different imaging technologies [155].

Kumar et al. reported the use of ORMOSIL nanoparticles for in vivo bioimaging, biodistribution, clearance, and toxicity studies with the use of multimodal methods. ORMOSIL nanoparticles with diameters of 20–25 nm were conjugated with DY776 (NIR fluorophores) and radiolabeled with ^124^I for optical and PET imaging in vivo. The biodistribution of nanoparticles was studied in non-tumor nude mice by optical fluorescence imaging and by measuring the radioactivity from harvested organs. The PET images showed a very similar pattern of uptake of the nanoparticles, as obtained from the fluorescence imaging of the mice. Fluorescence images in vivo showed data on the clearance of the DY776-conjugated ORMOSIL from a small batch of injected mice. DY776-conjugated ORMOSIL-injected mice initially showed liver and spleen accumulation over a period of 24 h, whereas maximum fluorescence was acquired from the skin at 72 h post-injection. The DY776 ORMOSIL facilitated optical bioimaging in the NIR window, with maximum tissue penetration of light and minimum background signal. Furthermore, the nanoparticle with DY776 and ^124^I allowed for bioimaging independent of tissue depth, as well as the more accurate quantification of accumulation of nanoparticles in various major organs in vivo [135].

Over the past decade, C dots have been investigated, for both image-guided surgical and therapeutic applications, by the Bradbury group and the Wiesner group. For clinical evaluation of tumors, a new generation of multimodal (PET-optical) C dot-bearing peptides that bind with various classes of receptors for tumor cells and/or aberrant intracellular targets have been reported [87,156,157]. Ultrasmall (<10 nm diameter) dual-modality optical and positron emission tomography silica nanoparticles (^124^I-cRGDY-PEG-Cy5-C dots), which integrate functional moieties within ultrasmall silica nanoparticles (3–10 nm), such as encapsulated Cy5 fluorescent dye, PEG stealth layer, and cRGDY peptide fragments with radioiodine, have been synthesized and characterized, demonstrating their long-term stability [157].

In vitro binding kinetics and targeting specificity of the number of ligands (6, 14, and 18) per particle have been evaluated in M21 cells by flow cytometry with the optical properties of cRGDY-PEG-Cy5-C dots. Enhanced uptake was observed with increasing particle concentrations. Eighteen ligands per particle gave the maximum cellular uptake. To demonstrate tumor-specific targeting, both M21 and M21-L tumor-bearing mice were intravenously injected with ^124^I-cRGDY-PEG-Cy5-C dots, which gave more favorable target-to-background ratios and imagery using a small animal PET system. The difference of surface ligand density was the key to determining biological activities, optimum binding affinity, cellular uptake, specificity, stability, and favorable pharmacokinetics in vitro and in vivo in melanoma models. The designed ultrasmall silica nanoparticles, which were functionalized with anti-human epidermal growth factor receptor 2 (HER2) single-chain variable fragments, exhibited high tumor-targeting efficiency and efficient renal clearance [87]. ^89^Zr-DFO-scFv-PEG-Cy5-C dots (DFO: deferoxamine) have been synthesized, which integrated five functional moieties within a single 6–7 nm silica nanoparticle: encapsulated Cy5 fluorescent dye, PEG stealth layer, ^89^Zr-chelated DFO, and anti-HER2 (scFv fragments). The tumor-targeting efficiency and uptake within HER2+ and HER2− xenograft models of ^89^Zr-DFO-scFv-PEG-Cy5-C dots have been investigated using optical imaging and PET imaging. The ^89^Zr-DFO-scFv-PEG-Cy5-C dots were systemically administered to non-tumor-bearing or BT-474 tumor-bearing mice, for assessment of biodistribution, radio stability, whole-body clearance, and HER2-targeted uptake in vivo. PET images showed dominant cardiac activity at 2  h of ^89^Zr-DFO-scFv-PEG-Cy5-C dots, indicating that the particles were largely confined to the blood pool. Cardiac activity concentrations gradually decreased over time. Hepatic uptake was also found to decrease from 2 to 72  h. Dominant bladder uptake was clearly observed in all three groups on both coronal and axial tomographic PET images at 2  h (Figure 4A–C). The uptake in muscle remained relatively constant over the imaging period. Similar trends were found for blood, liver, and muscle time–activity profiles, derived for two control groups: non-targeted (^89^Zr-DFO-Ctr/scFv-PEG-Cy5-C dots; BT-474 mice) and targeted (^89^Zr-DFO-scFv-PEG-Cy5-C dots; MDA-MB-231 mice). The penetration and distribution of ^89^Zr-DFO- scFv-PEG-Cy5-C dots were evaluated with ex vivo tumor tissue specimens by widefield fluorescence microscopy, immunohistochemical staining for HER2 expression, H&E staining, autoradiography, and confocal microscopy of tumor tissue specimens. The widefield fluorescence microscopy confirmed significant tissue penetration and diffusion of targeted particles throughout BT-474 specimens, which overlapped autoradiographic images. The functionalization of nanoparticles with antibodies was achieved for tumor targeting. Ultrasmall nanoparticles have broad utility, in particular for imaging various tumor tissues by suitably adapting the targeting fragment. The ultrasmall targeted nanotheranostic/nanotherapeutic platform possesses great potential as an imaging probe for tumor tissues and drug delivery vehicles.

## 5. Phototherapy

The therapeutic properties of light have been well-known for thousands of years. Phototherapy, including photodynamic therapy (PDT) and photohyperthermia therapy (PHT), mainly relying on phototherapeutic agents which target toxic reactive oxygen species (ROS) and heat to kill tumors, have attracted a good deal of attention, due to their non-invasive properties and negligible drug resistance. At this moment, PDT is being clinically tested for it use in oncology to treat cancers of the head and neck, brain, lung, pancreas, intraperitoneal cavity, breast, prostate, and skin [158]. PHOTOFRIN (dihematoporphyrin ether) is a first-generation photosensitizing agent used for the PDT of tumors and high-grade dysplasia in Barrett’s esophagus [159]. As for the currently reported silica-based phototherapy approaches, we describe two therapies in the following.

### 5.1. Photodynamic Therapy

PDT combined with nanoparticles has emerged as an alternative to chemotherapy or radiotherapy for the treatment of various diseases, including cancer [160,161,162,163]. Photosensitizers (PS) possess the properties to form ROS under light irradiation, leading to the destruction of cancer cells by apoptosis and necrosis. The PS, which accumulate in the tumor tissue, are often used in PDT therapies. Silica-based nanoparticles encapsulating PS in the silica matrix have emerged as a promising concept for applications in PDT [162]. Indeed, encapsulation should lead to the administration of the PS in a monomeric form without loss of activity. PS such as m-THPC [164], hypocrellin B [165], protoporphyrin IX (PpIX) [166], and methylene blue (MB) [167] have been encapsulated in silica nanoparticles (covalent or non-covalent), due to the EPR effect of silica nanoparticles for tumor tissues. PpIX silica nanoparticles have been investigated for effective use in PDT in vitro and in vivo models, in order to improve the role of biological factors in photodamage. PpIX silica nanoparticles are more efficient than free PpIX. A strong fluorescence signal of ROS generation colocalized with PpIX silica nanoparticles was correlated with 100% cell death [166].

Kohle et al. investigated the effect of the position of the PS in silica nanoparticles. Two types of ultrasmall poly(ethylene glycol)-coated (PEGylated) fluorescent core–shell silica nanoparticles with methylene blue derivate (MB2) were synthesized [167]. In Design 1, MB2 was encapsulated into the matrix of the silica core, while MB2 was grafted onto the silica core surface in between chains of the sterically stabilizing PEG corona in Design 2. Their singlet oxygen quantum yields, Φ_Δ_, with effective Φ_Δeff_ per particle reached 111 ± 3% and 161 ± 5% for Designs 1 and 2, respectively, substantially exceeding that of MB2 alone. Furthermore, the encapsulation of MB2 significantly improved PS photostability, while surface conjugation of MB2 diminished its photostability, compare with free MB2. Moreover, these particle designs allow for functionalization with a targeting peptide. The energy-transferring silica nanoparticles for two-photon-excited (TPE) PDT have been reported [168]. TPE silica nanoparticles have been designed, based on three main mechanisms: (1) TPE-electron transfer; (2) TPE-energy transfer; and (3) TPE-photolysis. TPE-silica nanoparticles can be applied to the diagnosis and therapy of cancer, with drastically lowered side effects, compared to those of chemotherapy, as well as for other diseases or in ultrasensitive bioimaging under the extremely low scattering of two-photon irradiation [169].

### 5.2. Photo Hyperthermia

Hyperthermia is a cancer treatment that kills cancer cells by taking advantage of their weakness to heat. Hyperthermia induces DNA double-strand breaks in cancer cells by warming them to 41.5–45.5 °C [170]. PHT is a technique in which nanoparticles are delivered to a target tumor tissue, convert laser irradiation into heat, and destroy malignant cells. This technique provides a promising strategy for cancer treatment, as it is non-invasive and can target specific cells to minimize harmful side effects. Silica nanoparticles are utilized by functionally fusing with various heat-generating materials. Silica nanoparticles can stabilize such materials, both thermodynamically and chemically [171].

Gold nanoparticles are the typical heat-generating material used in PHT. A silica core has been used to apply gold nanoparticles for PHT in the NIR region. Usually, the absorption of gold nanoparticles, such as nanospheres, is in the visible range. Gold-covered silica nanoparticles cause anisotropy in gold, shifting the gold plasmon band to longer wavelengths. Thus, the consequent gold nanoparticles possess strong absorption in the NIR region [172]. PHT using gold-silica nanoshell (silica core) composed of a silica nanoparticle coated with an ultra-thin gold layer has been reported. Intravenously injected polyethylene glycol (PEG)-coated nanoshells into a mouse tumor model with murine colon carcinoma cells (CT26.WT) was exposed to near-infrared laser light (808 nm) at its tumor site. As a result, the temperature increase in and damage to tumor tissue, along with inhibition of tumor growth, was observed, thus improving survival [173,174].

The technology of the combination of imaging and PHT has also progressed. Shan et al. presented a nanoparticle composed of NaYF_4_: Er^3+^, Yb^3+^ nanoparticle (UNP) cores surrounded by a silica shell doped with carbocyanine dye molecules that strongly absorb NIR. These materials combine light emission (from the upconverting core nanoparticles) with strong NIR absorption from the carbocyanine dyes in the silica shell, in order to allow for both optical imaging and photothermal treatment, respectively. Experiments with RAW264.7 showed that these particles are effective, in terms of up-take, and could be easily visualized by upconverting imaging using a 980 nm excitation source. Using the nanoparticles under 750 nm laser excitation indicated that 42% of RAW264.7 cells were killed [175].

In recent years, the combination of PHT with other therapies has been developed, due to the multifunctionalization of silica nanoparticles. Adjuvant therapies, such as a combination of chemotherapy and PHT, have been shown to have synergistic effects on the death of cancer cells. Nagesetti et al. synthesized ORMOSIL nanoparticles loaded with NIR dye IR820 and doxorubicin (DOX). This multifunctional ORMOSIL nanoparticle reported experiments that caused DOX release and temperature increase due to NIR irradiation in ovarian cancer cells (Skov-3), leading to cell death. The combination of DOX chemotherapy and PHT with laser irradiation promoted higher cancer cell death, compared to individual therapies. Additionally, the multifunctional ORMOSIL nanoparticle was able to generate high fluorescence for in vivo imaging. This nanoparticle could serve as a useful therapeutic probe for image-guided chemotherapy with adjuvant hyperthermia [176].

Iron oxide is already used for magnetic hyperthermia in clinical settings. The properties of iron oxide as a sensitizer for PHT have been recently reported [177,178]. One study evaluated the PHT effect of superparamagnetic iron oxide nanoparticles (SPION) and SPION encapsulated in silica (silica shell) nanoparticles (SPIONs-SIL) [179]. They compared the photothermal effects between SPION and SPIONs-SIL by NIR irradiation (808 nm laser) in an aqueous suspension. The temperature increase with SPIONs-SIL (45.7 °C) was higher than that with SPION (43.5 °C). The authors also reported that the temperature increase in these particles using PHT was higher than those using magnetic hyperthermia. These results demonstrate a great photothermal effect, which should be expected from the functional fusion of iron oxide and silica nanoparticles. PHT using hybrid silica nanoparticles combined with various functionalities has thus shown promising results and potentials.

## 6. Conclusions and Future Perspectives

Among various biomedical nanoparticles, including mesoporous silica nanoparticles, non-porous silica nanoparticles have demonstrated high potential for the realization of “photo-theranostic” techniques. Currently, non-porous silica nanoparticles are being actively developed for clinical applications. C dots have received approval from the U.S. Food and Drug Administration (FDA) for their first-in-human clinical trial as cancer-specific probes. The clinical trial revealed the effectiveness and safety of C dots as hybrid PET-optical imaging agents [156,180]. In addition, AGuIX^®^ nanoparticles made of silica and gadolinium have been developed as novel theranostics radiosensitizing nanoparticles. AGuIX^®^ has been accepted in clinical trials and is undergoing testing [181,182,183,184,185]. As an FDA-approved silica material, TheraSphere has been accepted for the selective internal radiation therapy of primary and metastatic hepatic malignancies [186,187,188,189,190,191]. These nano- and micro-sized clinically translated particles are non-porous, dense silica particles. Therefore, silica nanoparticles are highly promising materials in clinically translational “photo-theranostics”.

## Figures and Tables

**Figure 1 biomedicines-09-00073-f001:**
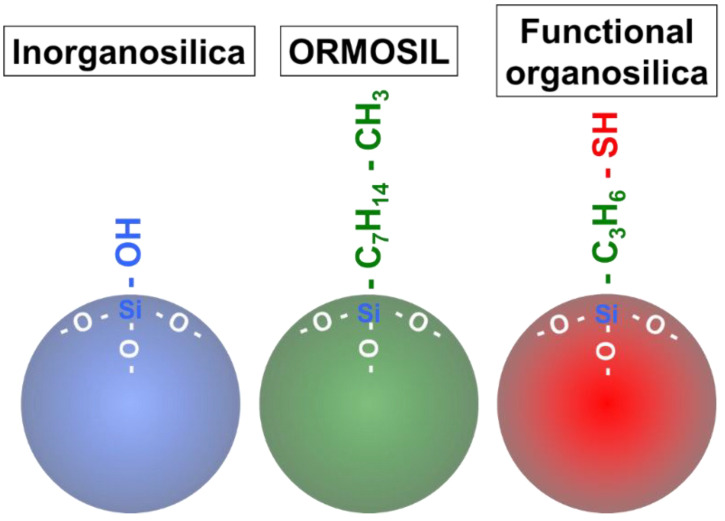
Schematic structure of three types of silica nanoparticles.

**Figure 2 biomedicines-09-00073-f002:**
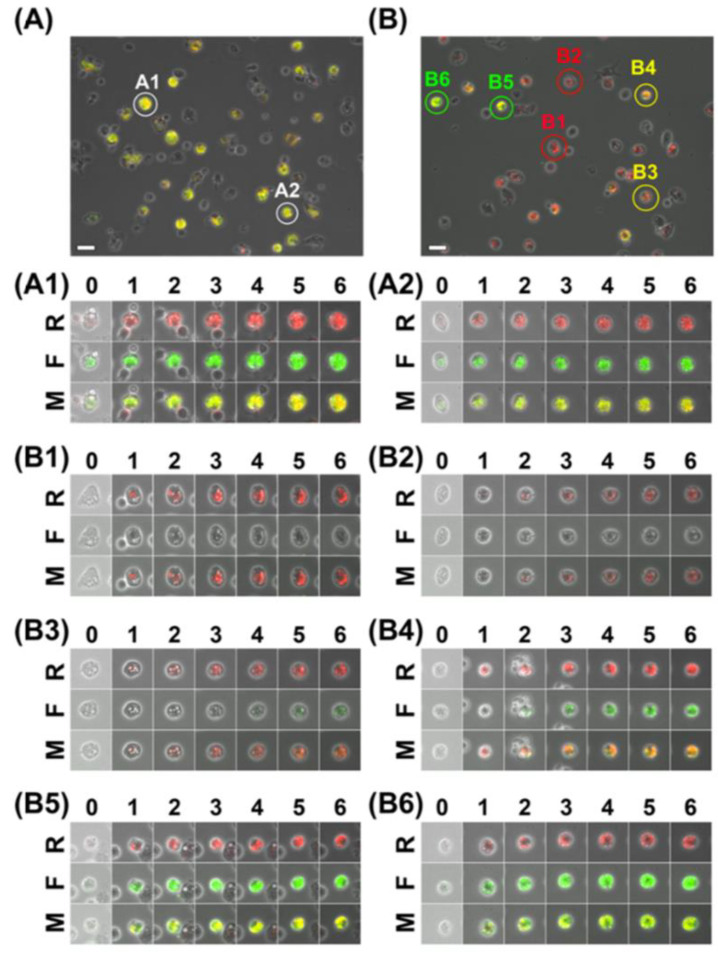
Single-cell dual-particle images showing macrophage uptake of fluorescent organosilica nanoparticles surface functionalized with PEG. The macrophages were incubated with a mixture of Flu and Rho (**A**) or with a mixture of Flu-PEG30K and Rho (**B**). The fluorescence images of Rub (R), Flu, or Flu-PEG30K (F) and merged images (M) of single cells are shown, from 0 to 6 h. The scale bars represent 40 μm. (**A1**,**A2**) Cells showed similar uptake of Flu and Rho. (**B1**,**B2**) Cells showed uptake of Rho and almost no uptake of Flu-PEG30K. (**B3**,**B4**) Cells showed uptake of Rho and a lower uptake of Flu-PEG30K. (**B5**,**B6**) Cells showed a similar uptake of Flu, Rho, and Flu-PEG30K. Reproduced with permission from Nakamura, M.; Hayashi, K.; Nakano, M.; Kanadani, T.; Miyamoto, K.; Kori, T.; Horikawa, K. Identification of polyethylene glycol-resistant macrophages on stealth imaging in vitro using fluorescent organosilica nanoparticles. *ACS Nano*
**2015**, *9*, 1058–1071 [127].

**Figure 3 biomedicines-09-00073-f003:**
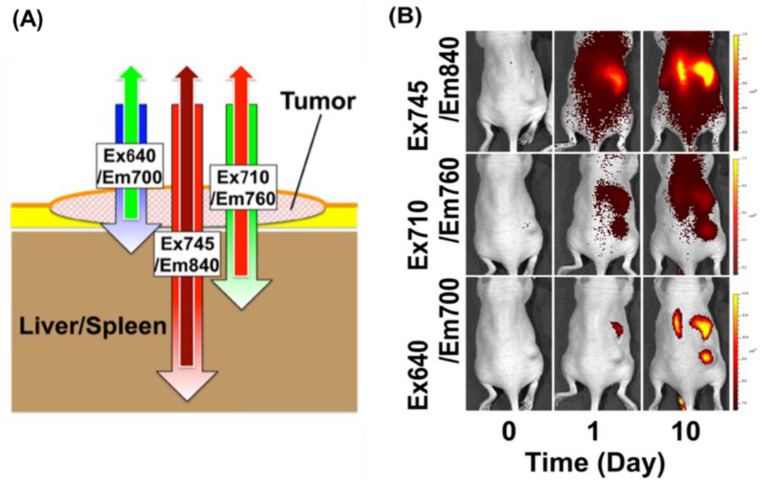
Depth-dependent NIR fluorescence in vivo imaging of a tumor. (**A**) A schematic illustration of depth-dependent NIR fluorescence imaging. Longer excitation and emission wavelengths, such as 745/840 nm (Ex745/Em850), can penetrate tissue very well and can detect particles in deeper sites, compared to shorter wavelengths (i.e., Ex710/Em760 and Ex640/Em700). (**B**) We administered totals of 2 mg (Day 1) and 6 mg (Day 10) of thiol-OS/IR820 into a mouse that had a subcutaneous xenograft tumor intravenously and observed the fluorescence using the in vivo imaging system under three excitation/emission wavelength conditions (Ex640/Em700 nm, Ex710/Em760 nm, and Ex745/Em840 nm). Reproduced with permission from Nakamura, M.; Hayashi, K.; Nakamura, J.; Mochizuki, C.; Murakami, T.; Miki, H.; Ozaki, S.; Abe, M. Near-Infrared Fluorescent Thiol-Organosilica Nanoparticles That are Functionalized with IR-820 and Their Applications for Long-Term Imaging of in Situ Labeled Cells and Depth-Dependent Tumor in Vivo Imaging. *Chem. Mater.*
**2020**, *32*, 7201–7214 [106].

**Figure 4 biomedicines-09-00073-f004:**
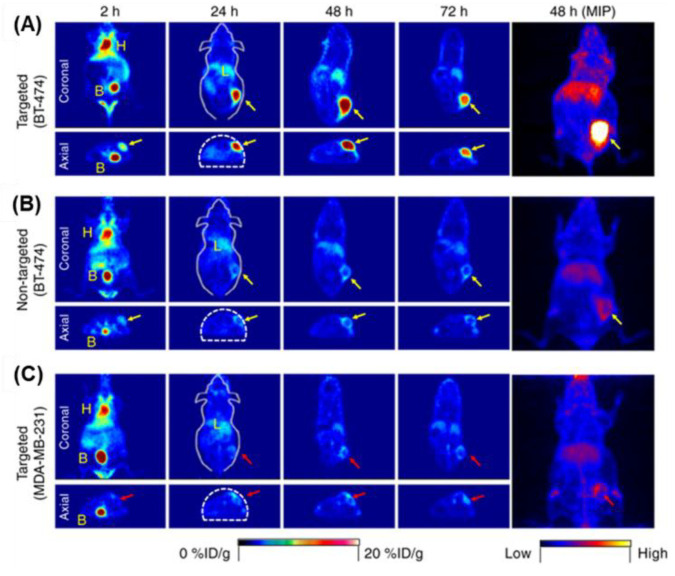
In vivo HER2-targeted PET imaging in xenograft breast cancer models. Serial coronal and axial tomographic PET images acquired at 2, 24, 48, and 72 h post intravenous injection of radiolabeled particle immunoconjugates in groups of tumor-bearing mice (N = 5 for each group), as follows: (**A**) targeted group, ^89^Zr-DFO-scFv-PEG-Cy5-C dots in BT-474 mice; (**B**) non-targeted group, ^89^Zr-DFO-Ctr/scFv-PEG-Cy5-C dots in BT-474 mice; and (**C**) targeted group, ^89^Zr-DFOscFv-PEG-Cy5-C dots in MDA-MB-231 mice. For each group, maximum intensity projection (MIP) images were also acquired at 48 h p.i. H, heart; B, bladder; L, liver. All BT-474 tumors are marked with yellow arrows, while all MDA-MB-231 tumors are marked with red arrows. Reproduced with permission from Chen, F.; Ma, K.; Madajewski, B.; Zhuang, L.; Zhang, L.; Rickert, K.; Marelli, M.; Yoo, B.; Turker, M.Z.; Overholtzer, M.; et al. Ultrasmall targeted nanoparticles with engineered antibody fragments for imaging detection of HER2-overexpressing breast cancer. *Nat. Commun.*
**2018**, *9*, 1–11 [87].

**Table 1 biomedicines-09-00073-t001:** Category of fluorescent materials for silica nanoparticles.

Fluorescent Materials			Reference
Organic	Fluorescent dye	Fluorescein isothiocyanate (FTIC), Rhodamine B, Eosin yellowish, Propidium iodide, Cyanine dyes (Cy3, Cy5, Indocyanine green (ICG), IRDye 800CW, IR820)	[60,61,74,79,80,81,82,83,84,85,86,87]
Inorganic	Rare earth	Eu, In	[88,89,90,91]
	Transition metal complex	tris(1,10-phenanthroline) ruthenium (II) chloride (Ru(phen)_3_^2+^), tris(2,2′-bipyridyl) dichlororuthenium(II) (Ru(bpy)_3_^2+^)	[92,93,94,95,96,97,98]
	Quantum dots	CdSe/ZnS, CdTe, CdSe/CdS/ZnS	[78,99,100,101,102]

## Data Availability

No new data were created or analyzed in this study. Data sharing is not applicable to this article.

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
