# Peer review of "Development of Non-Porous Silica Nanoparticles towards Cancer Photo-Theranostics"

_biomedicines, 2021, doi:10.3390/biomedicines9010073_

Round 1 (The peer review reports for the earlier submission)

Reviewer 1 Report

The manuscript "Development on nonporous silica nanoparticles toward photo-theranostics" by Chihiro Mochizuki et al. is a review of the synthesis and application of non-porous silica nanoparticles in nanomedicine with intended focus on photo-teranostics.

The article structure is designed in a mindful way; the covered literature is wide and recent enough. Unfortunately, the English of the article is not acceptable. The manuscript needs a profound revision by a native speaker who is also expert of the field – as the actual text contains not only grammatical errors, but also syntactical ones that at some points jeopardise the meaning of the text.

The article cannot be published in its current form.

Author Response

Thank you very much for your reviewing our manuscript (biomedicines-1036537). According to your helpful and valuable suggestions, we have revised the manuscript as follows:

Point 1: The article structure is designed in a mindful way; the covered literature is wide and recent enough. Unfortunately, the English of the article is not acceptable. The manuscript needs a profound revision by a native speaker who is also expert of the field – as the actual text contains not only grammatical errors, but also syntactical ones that at some points jeopardise the meaning of the text.

Response 1: Concerning the English in the manuscript, we used an English editing service by a native speaker and put the English proof.

Reviewer 2 Report

The manuscript entitled: Development of nonporous silica nanoparticles toward photo-theranostics' (Biomedicines - 1036537) by Michihiro Nakamura et al. review applications of silica nanoparticles in imaging and therapy of disease. In my opinion, this manuscript presents interesting content, but needs some improvements. Below are my comments:

  • in the title of the article it is stated that it concerns nanoparticles of nonporous silica however references are given to publications describing nanoparticles of mesoporous silica (e.g. 101, 107, 123, 145). Does this mean that the Authors do not consider the difference between these two types of silica nanoparticles??
  • the manuscript only provides examples of the use of silica nanoparticles for cancer diagnostics. Does this mean that in the case of other diseases these nanoparticles are not used in imaging and therapy? If silica nanoparticles are used in other diseases, it should be described or the Authors should indicate in the article title that it concerns the use of nanoparticles only for imaging and therapy of cancer;
  • why are there no other examples of using nanoparticles for photodiagnostics? It would be worth showing not only cell labeling but also taking into account other applications of silica nanoparticles, e.g. DNA – silica NP conjugates for multiplex DNA detection on microarrays;
  • the manuscript describes two types of nanoparticles that are treated as silica nanoparticles, the first of them are those with a silica core, while the second are hybrid nanoparticles in which the core made of a different material (Au or Fe3O4) is covered by silica shell. In my opinion Authors cannot treat these nanoparticles as the same group and call them "silica nanoparticles". This should be clarified by the Authors and the text of the manuscript should be rearranged (taking into account the different structure and composition of nanoparticles in separate chapters). In this way, it will be possible to show how the structure of nanoparticles influences their biomedical applications;
  • the content of the manuscript does not correspond to its title, theranostic is combination of two words: therapeutics and diagnostics, so Authors should show examples in which a diagnosis and application in therapy is acquired in one package (examples of nanoparticles for optical imaging allow cell labeling, however do not allow treatment of disease) or change the title to better relate to the content presented in the manuscript.

Taking into account above‑mentioned remarks, I did not recommend publication of this article in present form in Biomedicines. Authors should rewrite the article and re-submit it to the Biomedicines journal.

Author Response

Thank you very much for your reviewing our manuscript (biomedicines-1036537). According to your helpful and valuable suggestions, we have revised the manuscript as follows:

Point 1: in the title of the article it is stated that it concerns nanoparticles of nonporous silica however references are given to publications describing nanoparticles of mesoporous silica (e.g. 101, 107, 123, 145). Does this mean that the Authors do not consider the difference between these two types of silica nanoparticles??

Response 1: Concerning the distinguished nanoparticles of non-porous or mesoporous silica in the manuscript, we agree with this and have incorporated your suggestion throughout the manuscript. We deleted some of the references below to eliminate the confusion and focus on non-porous silica.

[deleted references]

>Tasciotti, E.; Godin, B.; Martinez, J.O.; Chiappini, C.; Bhavane, R.; Liu, X.; Ferrari, M. Near-infrared imaging method for the in vivo assessment of the biodistribution of nanoporous silicon particles. Mol. Imaging 2011, 10, 56–68.

>Dong, W.; Wen, J.; Li, Y.; Wang, C.; Sun, S.; Shang, D. Targeted antimicrobial peptide delivery in vivo to tumor with near infrared photoactivated mesoporous silica nanoparticles. Int. J. Pharm. 2020, 588, 119767.

>Huang, C.C.; Tsai, C.Y.; Sheu, H.S.; Chuang, K.Y.; Su, C.H.; Jeng, U.S.; Cheng, F.Y.; Su, C.H.; Lei, H.Y.; Yeh, C.S. Enhancing transversal relaxation for magnetite nanoparticles in mr imaging using Gd3+-chelated mesoporous silica shells. ACS Nano 2011, 5, 3905–3916.

>Cheng, S.H.; Hsieh, C.C.; Chen, N.T.; Chu, C.H.; Huang, C.M.; Chou, P.T.; Tseng, F.G.; Yang, C.S.; Mou, C.Y.; Lo, L.W. Well-defined mesoporous nanostructure modulates three-dimensional interface energy transfer for two-photon activated photodynamic therapy. Nano Today 2011, 6, 552–563.

Point 2: the manuscript only provides examples of the use of silica nanoparticles for cancer diagnostics. Does this mean that in the case of other diseases these nanoparticles are not used in imaging and therapy? If silica nanoparticles are used in other diseases, it should be described or the Authors should indicate in the article title that it concerns the use of nanoparticles only for imaging and therapy of cancer;

Response 2: Concerning mismatching the title and manuscript, we agree with this and have incorporated your suggestion throughout the manuscript. We revised the manuscript as below.

From [page 1, line 2]

"Development of Nonporous Silica Nanoparticles toward Photo-theranostics"

to [page 1, line 2]

"Development of Non-porous Silica Nanoparticles towards Cancer Photo-theranostics"

From [page 2, line 49]

"Recent research of nanomedicine is progressing toward the realization of the integration of diagnosis and therapy termed as "theranostics." The integration of optical imaging and phototherapy can be termed as "photo-theranostics" is one of the most important nanomedical goals. Recently many reviews of imaging and therapy, as well as "theranostics" using functional nanoparticles, have been published."

to [page 2, line 50]

"Recent research in nanomedicine has been progressing toward the realization of the integration of diagnosis and therapy, termed "theranostics". The integration of optical imaging and phototherapy, which can be termed "photo-theranostics", is one of the most important nanomedical goals at present. Many reviews focused on imaging and therapy, as well as "theranostics", using functional nanoparticles have been published, considering various diseases such as cancer, neurodegenerative [34,35], cardiovascular [36,37], and autoimmune (particularly rheumatoid arthritis) [38,39] diseases [11]. Cancer "theranostics" is one of the most active fields in nanomedicine."

Point 3: why are there no other examples of using nanoparticles for photodiagnostics? It would be worth showing not only cell labeling but also taking into account other applications of silica nanoparticles, e.g. DNA – silica NP conjugates for multiplex DNA detection on microarrays;

Response 3: Thank you for this suggestion. Concerning the lack of topics for silica application, we added sentences and some references to the manuscript and revised the manuscript as below.

To [page 5, line 165, postscript]

Fluorescent silica nanoparticles have been widely used in biological applications for bioassay [74–76] and imaging, due to their flexible chemistry and apparent biocompatibility.

Point 4: the manuscript describes two types of nanoparticles that are treated as silica nanoparticles, the first of them are those with a silica core, while the second are hybrid nanoparticles in which the core made of a different material (Au or Fe3O4) is covered by silica shell. In my opinion Authors cannot treat these nanoparticles as the same group and call them "silica nanoparticles". This should be clarified by the Authors and the text of the manuscript should be rearranged (taking into account the different structure and composition of nanoparticles in separate chapters). In this way, it will be possible to show how the structure of nanoparticles influences their biomedical applications;

Response 4: Concerning treating two types of nanoparticles (silica core and hybrid silica nanoparticles), We have accordingly changed the structure of the manuscript (see "3. Multi-functionalized Silica Nanoparticles") to emphasize this point. We revised the manuscript as below.

To [page 4, line 155, postscript]

3-2 Nanoparticles

Multifunctional silica nanoparticles have been designed to make use of the properties of combined materials; that is, organic molecules or inorganic materials such as novel metals and metal oxides. Among them, the multifunctionalization of silica nanoparticles with functional nanoparticles, such as gold nanoparticles or iron oxide nanoparticles, has garnered great attention for given biomedical applications. Multifunctional silica nanoparticles provide a significant impact not only in the biomedical field, but also for physical chemistry (e.g., for spectroscopy and magnetism). In these multifunctional nanoparticles, silica particles are used as cores or shells. In this section, we describe multifunctional silica nanoparticles, based on their structure.

3-2-1. Silica Shell Nanoparticles

Silica as a coating material has progressed to an unparalleled application for improving colloidal properties and abilities using core–shell rational designs, as well as benefitting from its manufacturing flexibility [108]. A number of reports have focused on the silica coating of inorganic nanoparticles by aqueous methods, such as the Stöber method [109,110], silane coupling agents, and sodium silicate; however, recently, water-in-oil (W/O) microemulsions [111] and the coating of polymeric aggregates with silica [112,113] have also been reported. Tuning of the relaxometry of γ-Fe2O3@SiO2 core−shell nanoparticles was carried out by changing the thickness of the covering silica layer. The control of silica shell thickness can, accordingly, be tuned, in order to allow a high response of the contrast agent [108].

3-2-2. Silica Core Nanoparticles

Silica nanoparticles fabricated from inorganic nanoparticles [114–121] provide attractive structures, due to the exposed inorganic nanoparticles which may facilitate magnetism and catalysis [118,119]. Monodisperse SiO2 nanoparticles (230 nm) were synthesized by the Stöber method and used as the core for Fe2O3 catalysts through hydrolysis precipitation. Fe2O3 nanoparticles were deposited to the silica surface through a hydroxyl bond [118]. Furthermore, gold nanoshells with a silica core have shown unique optical properties. The observed plasmon band in the NIR region relies on their center range and shell thickness by multipolar plasmon resonances [114–116]. The optical properties in the NIR region are considered to assess their potential for bioapplication into the biological window, as living cells and tissues have low light scattering and absorption in the NIR region. In particular, photothermal effects in the NIR region have gained attention for photo hyperthermia therapy, as discussed in a later section.

Point 5: the content of the manuscript does not correspond to its title, theranostic is combination of two words: therapeutics and diagnostics, so Authors should show examples in which a diagnosis and application in therapy is acquired in one package (examples of nanoparticles for optical imaging allow cell labeling, however do not allow treatment of disease) or change the title to better relate to the content presented in the manuscript.

Response 5: Concerning mismatching the title and manuscript, we revised the manuscript as below.

From [page 1, line 2]

"Development of Nonporous Silica Nanoparticles toward Photo-theranostics"

to [page 1, line 2]

"Development of Non-porous Silica Nanoparticles towards Cancer Photo-theranostics"

From [page 2, line 49]

"Recent research of nanomedicine is progressing toward the realization of the integration of diagnosis and therapy termed as "theranostics." The integration of optical imaging and phototherapy can be termed as "photo-theranostics" is one of the most important nanomedical goals. Recently many reviews of imaging and therapy, as well as "theranostics" using functional nanoparticles, have been published."

to [page 2, line 50]

"Recent research in nanomedicine has been progressing toward the realization of the integration of diagnosis and therapy, termed "theranostics". The integration of optical imaging and phototherapy, which can be termed "photo-theranostics", is one of the most important nanomedical goals at present. Many reviews focused on imaging and therapy, as well as "theranostics", using functional nanoparticles have been published, considering various diseases such as cancer, neurodegenerative [34,35], cardiovascular [36,37], and autoimmune (particularly rheumatoid arthritis) [38,39] diseases [11]. Cancer "theranostics" is one of the most active fields in nanomedicine."

Reviewer 3 Report

The review by Mochizuki et al addresses recent developments in nonporous silica nanoparticles towards application in imaging and phototherapy. The topic of the review is in line with the scope of the journal. The authors demonstrate that this is a timely and relevant review given the fact that it is a hectic field of research and no recent reviews exist. The manuscript is well structured, seminal and recent papers are acknowledged. I think the review has potential to be published but some issues need to be more carefully addressed.

1) I do find that the authors present in many aspects a very personal view of the topic. There is no mention of the use of ICG and IRdye 800CW in silica nanoparticles. ICG is the only FDA approved dye up till now and widely used in the clinic, and many diagnostic tools using IRdye 800CW are currently under preclinical trials. Thus, it is rather incomplete to discuss the application of cyanine dye loaded silica nanoparticles in imaging without a reference to those two dyes. IRdye 800CW in silica nanoparticles has been reported in Ma et al Chem. Mater. 2015, 27, 4119−4133, which is a seminal paper in the topic of the review and that was not duly cited. ICG has been also loaded in silica nanoparticles by Quan et al Talanta, 2012, 99, 387-393.

2) I would expect to see in such a focused review some systematic discussion about the size effect on particle extravasation, diffusion and internalization on tumor. The relationship between particle size and structure and pharmacokinetics is currently an ongoing discussion. Some researchers argue that small and ultra-small targeted particles are preferable for cancer therapy, whereas others argue that large particles relying on EPR effect are better suited because they can carry more dyes and drug. This topic should be specifically addressed.

3) Safety and biodegradability aspects of silica nanoparticles have also not bee conveniently discussed.

4) Finally, I found typos and grammatical errors that need to be carefully addressed.

L45: “being expanding”, instead of being expanded

L160: “was known as the day for used”, not clear

L214: “Cells labeled with dual fluorescent thiol-OS-QDs was”, was instead of were

L315 “nanoparticles was”, was instead of were

L353: “macrophages taken up thiol-OS/IR820 existing in the liver and spleen was also detected”, was instead of were.

L404: “nanoparticles  was  successfully”,  was instead of were

and the list could go on….

In ref 86 the author name is misspelled “Wiesnert” should be Wiesner

Author Response

Thank you very much for your reviewing our manuscript (biomedicines-1036537). According to your helpful and valuable suggestions, we have revised the manuscript as follows:

Point 1: I do find that the authors present in many aspects a very personal view of the topic. There is no mention of the use of ICG and IRdye 800CW in silica nanoparticles. ICG is the only FDA approved dye up till now and widely used in the clinic, and many diagnostic tools using IRdye 800CW are currently under preclinical trials. Thus, it is rather incomplete to discuss the application of cyanine dye loaded silica nanoparticles in imaging without a reference to those two dyes. IRdye 800CW in silica nanoparticles has been reported in Ma et al Chem. Mater. 2015, 27, 4119−4133, which is a seminal paper in the topic of the review and that was not duly cited. ICG has been also loaded in silica nanoparticles by Quan et al Talanta, 2012, 99, 387-393.

Response 1: Thank you for this suggestion. Concerning no mention of the use of ICG and IRdye 800CW in silica nanoparticles, we added sentences and some references to the manuscript and revised the manuscript as below.

From [page 4, line 175 [page 5, line 136]

Table 1. Category of fluorescence materials for silica nanoparticles

Fluorescence materials

Reference

Organic

Organic dye

FITC, Rhodamine B, Eosin yellowish, Cyanine dyes (Cy3, Cy5, IR820)

[54,55,64–70]

Inorganic

Rera earth

Eu, In

[71–74]

Transition metal complex

Ru complex

[75–81]

Quantum dots

CdSe/ZnS, CdTe, CdSe/CdS/ZnS

[63,82–85]

to [page 5, line 175]

Table 1. Category of fluorescent materials for silica nanoparticles

Fluorescent materials

Reference

Organic

Fluorescent dye

Fluorescein isothiocyanate (FTIC), Rhodamine B, Eosin yellowish, Propidium iodide, Cyanine dyes (Cy3, Cy5, Indocyanine green (ICG), IRDye 800CW, IR820)

[60,61,74,79–87]

Inorganic

Rare earth

Eu, In

[88–91]

Transition metal complex

tris(1,10-phenanthroline) ruthenium (II) chloride (Ru(phen)32+), tris(2,2′-bipyridyl) dichlororuthenium(II) (Ru(bpy)32+)

[92–98]

Quantum dots

CdSe/ZnS, CdTe, CdSe/CdS/ZnS

[78,99–102]

From [page 5, line 162]

"Cyanine dyes are cationic molecules in which two heterocyclic units are joined by a polyene chain [68]. Specially Cy3, Cy5, and IR820 are known as fluorescence probes for bioimaging [69,70]."

to [page 6, line 201]

"Cyanine dyes are cationic molecules in which two heterocyclic units are joined by a polyene chain [85]. Cy3, Cy5, indocyanine green (ICG), IRdye 800CW, and IR820 are well-known fluorescence probes for bioimaging [74,79,86,87]. Among them, ICG is an FDA-approved near-infrared (NIR) fluorescent dye used in clinical imaging; more recently, IRdye 800CW is currently undergoing preclinical trials. ICG molecules ion-paired with a cationic polymer polyethylenimine (PEI) have been encapsulated into a silica nanoparticle using the Stöber method. PEI acted to prevent the aggregation of ICG in the silica and reduced the self-quenching of fluorescence [79]."

Point 2: I would expect to see in such a focused review some systematic discussion about the size effect on particle extravasation, diffusion and internalization on tumor. The relationship between particle size and structure and pharmacokinetics is currently an ongoing discussion. Some researchers argue that small and ultra-small targeted particles are preferable for cancer therapy, whereas others argue that large particles relying on EPR effect are better suited because they can carry more dyes and drug. This topic should be specifically addressed.

Response 2: Concerning the size effect of silica nanoparticles, we added some references to the manuscript and revised the manuscript as below.

To [page 3, line 88, postscript]

"The application of silica nanoparticles in the biomedical field can provide great advantages. Specially size-controlled nanoparticles have useful behavior in the body, compared to small molecules. Small molecules disperse to different organs, tissues, cells, and intracellular organelles in the body. On the other hand, nanoparticles have relatively long-term retention time in the body, as appropriately sized nanoparticles can accumulate in tumor tissue significantly better than in ordinary tissues; this phenomenon is known as the enhanced permeability and retention (EPR) effect [62,63]. Various size silica particles with a narrow size distribution can be prepared, which have been applied to study nano-sized effects. The appropriate size of nanoparticles in biodistribution for cancer therapy is currently an ongoing discussion."

Point 3: Safety and biodegradability aspects of silica nanoparticles have also not bee conveniently discussed.

Response 3: Concerning no mention of the Safety and biodegradability aspects of silica nanoparticles, we added some references to the manuscript and revised the manuscript as below.

To [page 3, line 96, postscript]

"Furthermore, the degradability and clearance of nanoparticles are essential phenomena for biomedical uses. Both biodegradable and renal clearable nanoparticles are known to cross the globular filtration obstruction of the kidneys, in order to be securely discharged. Silica nanoparticles have a fast hydrolytic rate and dissolve, over time, into water-soluble silicic acid, which is excreted in the urine [64,65]. The toxicity of the silica particles in in vivo systems has also been examined systematically [66,67]. The acute and sub-chronic toxicities of 50 nm and 500 nm non-porous silica nanoparticles and 500 nm mesoporous silica nanoparticles have been investigated by single-dose intravenous injection, where 50 nm non-porous silica nanoparticles and 500 nm mesoporous silica nanoparticles showed less sub-chronic toxicity [68]."

Point 4:  Finally, I found typos and grammatical errors that need to be carefully addressed.

L45: "being expanding", instead of being expanded

L160: "was known as the day for used", not clear

L214: "Cells labeled with dual fluorescent thiol-OS-QDs was", was instead of were

L315 "nanoparticles was", was instead of were

L353: "macrophages taken up thiol-OS/IR820 existing in the liver and spleen was also detected", was instead of were.

L404: "nanoparticles  was  successfully",  was instead of were

and the list could go on….

In ref 86 the author name is misspelled "Wiesnert" should be Wiesner

Response 4: Thank you for pointing this out. Concerning typos and grammatical errors in the manuscript, we used an English editing service by a native speaker and put the English proof. we revised the manuscript as below.

From L45: "being expanding", instead of being expanded

to L47: “been being expanded”

From L160: "was known as the day for used", not clear

to L199: Eosin yellowish, “which is the form of eosin used most often as a histologic stain”,

From L214: "Cells labeled with dual fluorescent thiol-OS-QDs was", was instead of were

to L291: “Cells labeled with dual-fluorescent thiol-OS-QDs have been functionalized” with

From L315 "nanoparticles was", was instead of were

to L392: IR-820-doped ORMOSIL “nanoparticles have been” applied to

From L353: "macrophages taken up thiol-OS/IR820 existing in the liver and spleen was also detected", was instead of were.

to L431: while “macrophages which had taken up thiol-OS/IR820 in the liver and spleen were also detected”

From L404: "nanoparticles  was  successfully",  was instead of were

to L486: “nanoparticles were successfully”

From (In ref 86) the author name is misspelled "Wiesnert" should be Wiesner

to (ref 103) Ow, H.; Larson, D.R.; Srivastava, M.; Baird, B.A.; Webb, W.W.; Wiesner, U. Bright and stable core-shell fluorescent silica nanoparticles. Nano Lett. 2005, 5, 113–117.

Round 2 (The peer review reports for the current submission)

Reviewer 1 Report

 The authors have addressed all the comments appropriately. The quality of the manuscript has been improved in this revised version.

I recommend accepting the manuscript in its present form

Reviewer 2 Report

I accept the revised version of this manuscript.

Reviewer 3 Report

The English of the article is definitely improved.